# Modified Time-Frequency Marginal Features for Detection of Seizures in Newborns

**DOI:** 10.3390/s22083036

**Published:** 2022-04-15

**Authors:** Nabeel Ali Khan, Sadiq Ali, Kwonhue Choi

**Affiliations:** 1Faculty of Engineering & IT, Foundation University Islamabad, Islamabad 46000, Pakistan; nabeel.ali@fui.edu.pk; 2Department of Electrical Engineering, University of Engineering and Technology, Peshawar 25000, Pakistan; sadiqali@uetpeshawar.edu.pk; 3Department of Information and Communicaiton, Yeungnam University, Gyeongsan 38541, Korea

**Keywords:** time frequency, EEG, newborns, seizure, classification, detection

## Abstract

The timely detection of seizure activity in the case of newborns can help save lives. Clinical signs of seizures in newborns are difficult to observe, so, in this study, we propose an automated method of detecting seizures in newborns using multi-channel electroencephalogram (EEG) recording acquired from 36 newborns admitted to Royal Women’s Hospital, Brisbane, Australia. A novel set of time-frequency marginal features are defined to detect seizure activity in newborns. The proposed set is based on the observation that EEG seizure signals appear either as a train of spikes or as a summation of frequency-modulated chirps with slow variation in the instantaneous frequency curve. The proposed set of features is obtained by extracting the time-frequency (TF) signature of seizure spikes and frequency-modulated chirps by exploiting the direction of ridges in the TF plane. Based on extracted TF signature of spikes, the modified time-marginal is computed whereas based on the extracted TF signature of frequency-modulated chirps, the modified frequency-marginal is computed. It is demonstrated that features extracted from the modified time-domain marginal and frequency-domain marginal in combination with TF statistical and frequency-related features lead to better accuracy than the existing TF signal classification method, i.e., the proposed method achieves an F1 score of 70.93% which is 5% greater than the existing method.

## 1. Introduction

Timely detection of seizures in newborns can help to save lives [1]. Clinical signs of seizures in newborns are difficult to observe, so, in this study, we propose an automated method of detecting seizures in newborns using multi-channel electroencephalogram (EEG) recording. Seizure detection usually requires monitoring of electroencephalogram (EEG) signals by an expert neurophysiologist. So, there is a need to develop an automated seizure detection algorithm. Machine learning based seizure detection algorithms usually involve the extraction of features from signal representation followed by training of a classifier signal [2,3]. The performance of such methods mainly depends on the discriminatory performance of extracted features. For feature extraction, the following forms of signal representations can be used: (a) time-domain, (b) frequency-domain, and (c) joint time-frequency (TF) domain [4,5,6,7,8]. It has been demonstrated in earlier studies that extracting features from TF features generally leads to better performance than time-domain features or frequency domain features [9,10]. One alternative to extracting features from joint TF representations is to first employ signal decomposition methods such as empirical mode decomposition to decompose signals into their components and then extract features from each component separately [1,11,12].

Recently, deep learning methods are employed for the automated detection of seizures. These methods do not require domain knowledge of an analyst to define features; rather, deep learning architectures automatically learn features from signal representation [13,14,15,16,17,18]. However, these methods usually require the availability of large databases.

We propose a novel approach to extracting features from joint TF representation. The proposed method is based on the observation that seizures either appear as a train of spikes or a summation of piece-wise linear frequency modulated chirps with slow variation in instantaneous frequency [6]. The TF signature of such signals is composed of ridges that are either aligned to the time axis or to the frequency axis [19,20]. The mathematical representation for a class of seizures that are modeled as a train of impulses is given as:(1)$$x\left(t\right)=\sum_{n=1}^{N}\delta (t-t_{n})$$
and the mathematical modulation of EEG seizures modeled as a summation of frequency-modulated chirps is given as:(2)$$x\left(t\right)=\sum_{n=1}^{N}\cos(2\pi f_{0}\left(t\right)tn)$$
where $f_{o}\left(t\right)$ is the instantaneous frequency of fundamental component. It is usually modeled as a piece-wise linear frequency modulated chirp [8]. To exploit this information, we extract features (a) from the modified time-marginal that extracts the time-domain signature of only spikes, (b) the modified frequency-marginal that extracts frequency-domain signatures of frequency-modulated chirps, and (c) from filtered TF representation that extracts the TF signature of seizure signals. Experimental results indicate that the proposed method achieves better performance than a state of the art in terms of F1-score. The proposed method achieves an F1-score of 0.709 whereas the F1-score of the existing method is 0.65.

The organization of the remainder of the article is as follows. Section 2 presents details of the methodology for extracting the proposed features for seizure detection. To assess the performance of the seizure detection scheme, numerical results are presented in Section 3 and Section 4 concludes the article.

## 2. Methodology

Multi-channel EEG signals are acquired and signals from all channels are averaged to get the signal x(t). The remaining process is performed on the averaged signal. The proposed system is illustrated in Figure 1 and the main steps are explained in the following sections.

### 2.1. Time-Frequency Analysis of EEG Signals

EEG signals, because of their non-stationary characteristics, are best analyzed in the joint TF domain using TF representations. Previous studies have shown that analyzing seizure signals in the joint TF domain reveals important features that can be used for their classification [21,22]. TF representations are generally classified into linear and quadratic classes [23]. We have employed quadratic TF representations because they represent the distribution of signal energy in the TF domain [23].

Wigner Ville distribution (WVD) is a core TF distribution of quadratic class defined as [24]:(3)$$W\left(t,f\right)=\int z\left(t+\frac{\tau }{2}\right)z^{*}\left(t-\frac{\tau }{2}\right)e^{-j2\pi f\tau }d\tau$$
where z(t) is the analytic associate of a real-life signal x(t). The WVD has high energy concentration for mono-component linear frequency modulated signals but it produces cross-terms for multi-component and non-linear frequency modulated signals [23]. The cross-terms in the WVD are typically reduced by convolving them with signal-dependent or independent kernels [25].

In our earlier studies, we demonstrated that an effective method to reduce cross-terms in WVD is through the application of signal-dependent adaptive directional kernel whose principal direction is adjusted at each TF point. TFDs obtained as a result of adaptive directional smoothing are called adaptive directional time-frequency distributions (ADTFDs) [26].
(4)$$\rho \left(t,f\right)=W\left(t,f\right)\underset{t}{*}\underset{f}{*}\gamma_{\theta }\left(t,f\right),$$
where $\gamma_{\theta }\left(t,f\right)$ is given as:(5)$$\gamma_{\theta }\left(t,f\right)=\frac{ab}{2\pi }e^{-a^{2}t_{\theta }^{2}-b^{2}f_{\theta }^{2}}\left(1-a^{2}k_{\theta }^{2}\right)$$
where $t_{\theta }=t\cos\left(\theta \right)+f\sin\left(\theta \right)$ and $f_{\theta }=f\cos\left(\theta \right)-t\sin\left(\theta \right)$, θ is the direction of $\gamma_{\theta }\left(t,f\right)$, *a* determines the extent of smoothing along the major axis, and parameter *b* determines the extent. We adjust the direction of $\gamma_{\theta }\left(t,f\right)$ at each TF point to align its principal direction with the major axis of ridges. The direction of principal axis of ridges is estimated as [26]:(6)$$\theta \left(t,f\right)=\underset{\theta }{\arg\max}\left|\left|\rho (t,f)\right|\underset{f}{*}\underset{f}{*}\gamma_{\theta }\left(t,f\right)\right|,$$
where θ is varied from $0^{\circ }$ degree to $180^{\circ }$ degree with the step size of $3^{\circ }$.

EEG seizure signals are modeled as either train of impulses or as the sum of harmonically related frequency modulated chirps, whereas background appears as noise. To enhance discriminatory characteristics of seizures, EEG signals are first pre-processed through differentiator filter [27] as: $\frac{d}{dt}x\left(t\right)$. This differentiation implementation can be implemented digitally as x(n)−x(n−1). Let us analyze two types of EEG seizure signals and EEG background signals in the joint TF domain using the ADTFD. Figure 2a illustrates a TF representation of EEG seizure signal with spiky characteristics, Figure 2b illustrates TF representation of seizure signal composed of frequency-modulated chirps, and Figure 2c illustrates TF representation of non-seizure signal.

It is observed that the TF signature of EEG seizures is composed of ridges that are either aligned to the time axis due to spikes or aligned to the frequency axis due to frequency-modulated chirps. However, in the case of background ridges, they can follow any random direction.

### 2.2. Modified Time-Marginal and Frequency-Marginal Features

In this study, we develop a novel approach to compute modified time-marginal and modified frequency marginal for extracting features from EEG signals. From TFDs, that satisfy marginal properties, time-marginal can be estimated from TFD as [23]:(7)$${\left|z(t)\right|}^{2}=\int \rho \left(t,f\right)df$$
or similarly, frequency-marginal can be estimated as:(8)$${\left|z(f)\right|}^{2}=\int \rho \left(t,f\right)dt$$

We define modified marginals to extract the time-domain signature of epileptic spikes or frequency-domain signatures of frequency-modulated chirps.

#### 2.2.1. Time Marginal Feature

Let us define a TF distribution, i.e., $\left(\rho_{t}\left(t,f\right)\right)$, that preserves only those ridges that are aligned to the frequency axis and has 0 value for all other TF points [28].
$$\begin{array}{cc} \rho_{t}\left(t,f\right)= & \rho \left(t,f\right)\;\;80^{\circ }<\theta \left(t,f\right)<100^{\circ }\\ \rho_{t}\left(t,f\right)= & 0\;\;\mathrm{otherwise} \end{array}$$

The time-domain signature of spikes can then be extracted from $\rho_{t}\left(t,f\right)$:(9)$$s\left(t\right)=\int \rho_{t}\left(t,f\right)df,$$

Let us consider a TF signature of EEG seizure signal with spiky characteristics as shown in Figure 3a. The TF signature of spikes is extracted by $\rho_{t}\left(t,f\right)$ as shown in Figure 3b. The time-domain marginals obtained from ρ(t,f) and $\rho_{t}\left(t,f\right)$ are illustrated in Figure 3c. Note that time-marginal obtained from ρ(t,f) contains energy from both spike and tone components whereas the time-marginal obtained from $\rho_{t}\left(t,f\right)$ only has energy from spikes.

Note that s(t) only contains spike components. We estimate the features from time-domain marginal of s(t) by dividing them into 4 non-overlapping equal segments and then estimating energy in each segment as:(10)$$T_{k}=\int_{2k}^{2(k+1)}s\left(t\right)dt,$$
where k=0,1,2,3.

#### 2.2.2. Frequency Marginal Feature

We adopt a similar approach to extract modified frequency-marginal. Let us define a TF representation that preserves ridges aligned with time-axis
$$\begin{array}{cc} \rho_{f}\left(t,f\right)= & \rho \left(t,f\right)\;\;\theta \left(t,f\right)<10^{\circ }\;\mathrm{or}\;\theta \left(t,f\right)>170^{\circ }\\ \rho_{f}\left(t,f\right)= & 0\;\;\;\mathrm{otherwise} \end{array}$$

We extract the frequency-domain marginal as:(11)$$s\left(f\right)=\int \rho_{f}\left(t,f\right)dt$$

We define features from modified frequency-domain marginal, i.e., s(f), by estimating energy from the sub-bands (0 Hz to 1 Hz), (1 Hz to 2 Hz), (2 Hz to 3 Hz), (3 Hz to 4 Hz), (4 Hz to 8 Hz), and (8 Hz to 16 Hz). The modified frequency marginal features are given in Table 1.

### 2.3. Time-Frequency Statistical and Time-Frequency Signal Features

Let us define the TFD that preserves ridges that are either aligned to the time axis or the ridges that are aligned along the frequency axis [20], i.e.,
(12)$$\rho_{\mathrm{FIL}}\left(t,f\right)=\rho_{t}\left(t,f\right)+\rho_{f}\left(t,f\right).$$

Figure 4 illustrates $\rho_{\mathrm{FIL}}\left(t,f\right)$ and ρ(t,f) for both background and seizure signals. It is observed that for seizure signals $\rho_{\mathrm{FIL}}\left(t,f\right)\approx \rho \left(t,f\right)$. However, for EEG background, $\rho_{\mathrm{FIL}}\left(t,f\right)$ is significantly different from ρ(t,f).

Based on this observation, we employ $\rho_{\mathrm{FIL}}\left(t,f\right)$ instead of ρ(t,f) for extracting TF statistical [9] and TF signal [9] related features. The list of features along with their respective formula are given in Table 2. The importance of each feature in the process of classification is found based on the decrease in accuracy when that feature is removed from the classification. The findings are presented in Section 3.3.

## 3. Experimental Results

### 3.1. Database Description

We have used a multi-channel newborn EEG database that is described in [4,29,30] and was made publicly available along with relevant codes at https://github.com/ElsevierSoftwareX/SOFTX-D-17-00059 (accessed on 1 November 2019). The database was acquired from 36 newborns using a 10–20 international electrode placement system at sampling frequency of 256 Hz. The marking of the channels is illustrated in Figure 5, which shows the possible placements of the electrodes. For each location, the first letter identifies the lobe whereas the second number or letter identifies the hemisphere location. The letters F, T, C, P, and O stand for Frontal, Temporal, Central, Parietal, and Occipital. The right hemisphere is represented by even numbers whereas odd numbers represent the left hemisphere. The z refers to an electrode placed on the mid-line. The signals were acquired at Royal Women’s Hospital Brisbane by an expert pediatric neurologist. The database was annotated for seizure, artifacts, and normal background patterns. Ethical approval was given by Royal Brisbane and Women’s Hospital (RBWH), Medical Research Ethics Committee (MREC), Brisbane, Australia [31]. The durations of background, seizures, and artifacts are 6 h, 5.16 h, and 5.04 h, respectively. EEG signals are segmented into non-overlapping segments of 8 s. The recorded dataset was divided into non-overlapping segments of duration 8-s. This database contains a total of 7290 segments including 2332 segments of seizures.

Each segment was labeled as 1 if seizure activity was observed and 0 if no seizure activity was observed. We analyzed the dataset and observed that approximately 78 percent seizure activity takes place below 4 Hz, 15 percent in between 4 Hz and 8 Hz, and 5.5 percent from 8 Hz to 12 Hz. Based on these observations, the signal is downsampled to 32 Hz to reduce the computational load. A snapshot of multi-channel EEG seizure signals is given in Figure 6.

### 3.2. Results and Discussion

In earlier studies, two alternate approaches of feature extraction from multi-channel EEG signals were proposed. The first approach converts multi-channel EEG recordings into the single channel through spatially averaging whereas the second approach converts extract features from each channel separately and then finally spatially averages the extracted features. In this study, we adopted the former approach to reduce computational costs. Illustrations of averaged and downsampled EEG signals just before the start of seizure signal and just after seizure signal are shown in Figure 7.

Let us now apply the proposed methodology to detect seizures in newborns. Cross-validation is performed using the leave-one-out approach where testing is performed by using EEG recordings of one patient while the EEG recordings of the remaining patients are used for training. This process is repeated until all patients are tested. We use equal examples of seizure and non-seizure segments from the training set. In order to reduce false positives, we first apply a 5-tap median filter, i.e., a given segment is classified as a seizure segment if a seizure appears in at least three out of five segments. The application of this median filter removes seizures appearing at the boundaries, thus reducing the overall sensitivity of the system. To improve the sensitivity, we extend the boundary by 8 s at each end. An illustration of the detected vs. original seizure using the balanced dataset is given in Figure 8.

For the same EEG recording, Figure 9 illustrates the plot of extracted features.

### 3.3. Discussion

Let us compare the performance of the proposed ADTFD-based TF features with the large set of TF features that include TF signal related features, TF image related features, TF Harlic features, and time-scale features proposed in [4]. This large set of features is extracted using Compact Kernel TFD (CKD), Spectrogram (SPEC), and Extended modified B distribution (EMBD). We have used F1-measure, sensitivity, total accuracy, specificity, and precision as criteria for performance comparison. The experimental results given in Table 3 indicate that the F1-score obtained by the proposed method is 70.93%, which is 5% greater than the results obtained by the next best method.

The importance of each feature in the random forest classifier is given in Figure 10. The importance of each feature is computed based on the decrease in accuracy when that feature is removed from the classification.

Experimental results indicate that the proposed approach of exploiting direction of ridges for extracting features from modified time-marginal, modified frequency-marginal, and filtered TF representation has significantly improved the classification accuracy and F1-score, e.g., the proposed method achieves the F1 score of 70.9%, which is 5% greater than the F1-score achieved by the next best method. These improved results are important as timely detection of seizures in newborns through an automated classification system can help save lives. The proposed approach is based on the observation that seizures either appear as a train of impulses or as a summation of harmonically related frequency modulated chirps, whereas in the case of background, ridges can follow any random directions. Based on these observations, a novel TF pre-processing method is developed where only those ridges that are either parallel to the time axis or parallel to the frequency axis are retained to extract TF signatures of seizures. The extracted TF signature is used to define novel features that lead to better accuracy than existing TF features. The proposed approach for the detection of seizure activity by extracting a signal of interest in the TF domain is not limited to the detection of seizures but can be adapted for the detection of abnormalities in a wide range of physiological signals, such as electrocardiogram signals.

The limitations of this study are as follows:High computational cost: The proposed approach is computationally expensive as it requires the computation of ADTFD with complexity of $O(N^{2}\log\left(N\right)+RM^{2}N^{2})$, where *M* is the size of the smoothing filter, *N* is the number of samples in a given segment, and *R* is the number of quantization levels. Note that the complexity of quadratic TFDs used in the earlier study was $O(N^{2}\log\left(N\right))$.For detection of seizures in multi-channel recordings, the better classification results are obtained by first extracting features from each channel separately and then reducing the dimension of features through dimensionality reduction approaches as compared to first combining all channels through averaging followed by feature extraction. Due to the high computational cost of the proposed method, we first combined all channels through averaging and then extracted features from the single combined channel. In the future, we will further improve the performance of our method through feature averaging.In this study, we used the EEG segment of 8-s duration sampled at 32 Hz for the detection of seizure activity. However, the results can be improved by using longer segments of EEG segments.Artifact detection and removal: Sometimes the machine learning algorithm wrongly detects artifacts as EEG seizures; this causes false positives. A number of artifact detection and removal systems have been developed. We can improve the performance of the proposed method by integrating artifact detection and removal system as a preprocessing step.

## 4. Conclusions

A novel set of TF features are defined for the classification of EEG signals. The proposed set of features is obtained by defining a TFD that extracts the TF signatures of seizures based on the direction of ridges; e.g., $\rho_{\mathrm{FIL}}\left(t,f\right)$, $\rho_{t}\left(t,f\right)$, and $\rho_{f}\left(t,f\right)$. Modified time-marginal features are extracted by projecting the TF signature of seizures with spike characteristics to the time axis and modified frequency-marginal features are extracted by projecting the TF signature of seizures composed of frequency-modulated chirps to the frequency axis. Features from both modified time-marginal and modified frequency-marginals are extracted in addition to TF statistical features, energy concentration measure, TF flux, and TF flatness, which are extracted from filtered TF representation obtained by preserving ridges aligned to the time axis or frequency axis, i.e., $\rho_{\mathrm{FIL}}\left(t,f\right)$. Experimental results indicate that the proposed set of TF features leads to better performance by achieving an F1-score of 70.93%, which is 5% greater than the existing EEG seizure detection algorithm based on TF feature extraction. In this study, we used segments of 8-second duration for feature extraction and classification. As part of the feature study, we will explore larger segments of seizures to improve the classification accuracy. In addition, an automated integrating artifact detection and removal system will be integrated as a pre-processing step to minimize false positives.

## Figures and Tables

**Figure 1 sensors-22-03036-f001:**
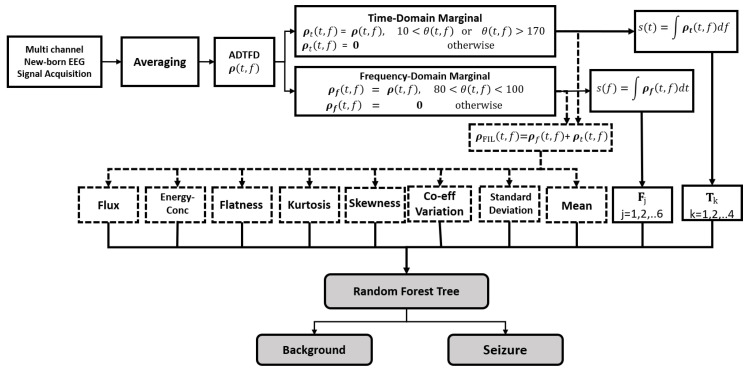
Illustration of the proposed system for detection of seizures in newborn EEG signals.

**Figure 2 sensors-22-03036-f002:**
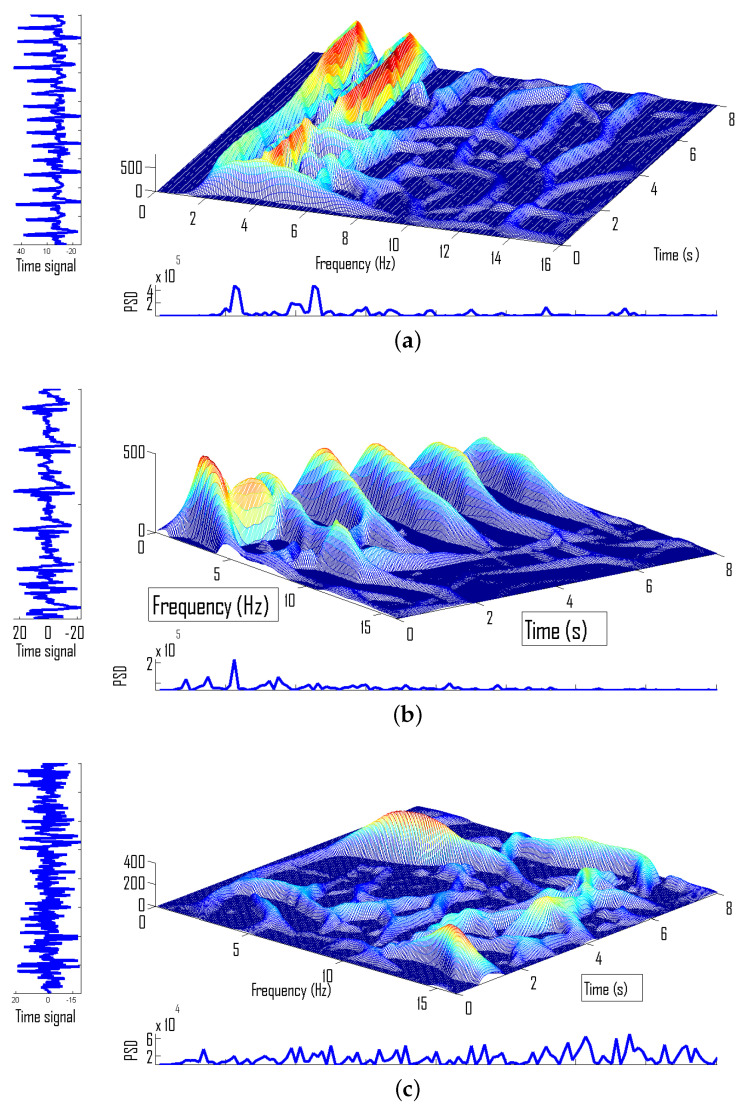
(**a**) EEG seizure composed of frequency modulated chirps, (**b**) EEG seizure composed of spike, (**c**) EEG background.

**Figure 3 sensors-22-03036-f003:**
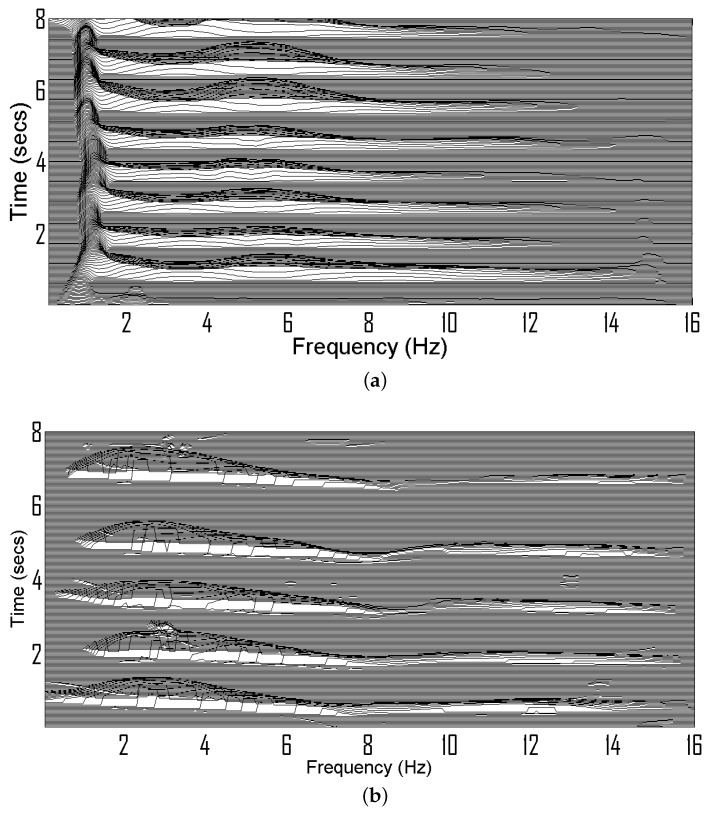
(**a**) ρ(t,f) of EEG seizure composed of tone and spike, (**b**) $\rho_{t}\left(t,f\right)$ of the EEG seizure signal, (**c**) time-domain marginal.

**Figure 4 sensors-22-03036-f004:**
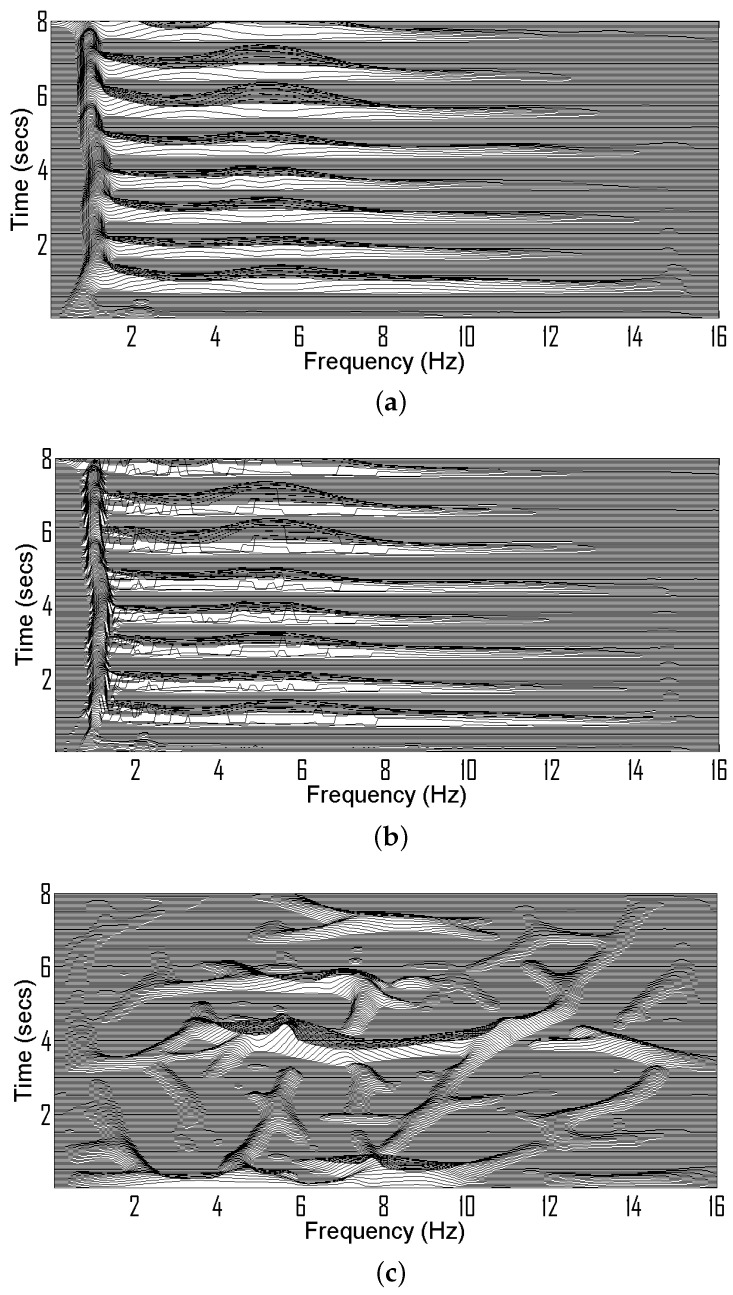
(**a**) TF signature of EEG seizure signal composed of summation of spikes and low frequency component, (**b**) Extracted TF signature of seizure signal obtained by $\rho_{\mathrm{FIL}}\left(t,f\right)$. (**c**) TF signature of EEG background, (**d**) TF signature of EEG background obtained by $\rho_{\mathrm{FIL}}\left(t,f\right)$.

**Figure 5 sensors-22-03036-f005:**
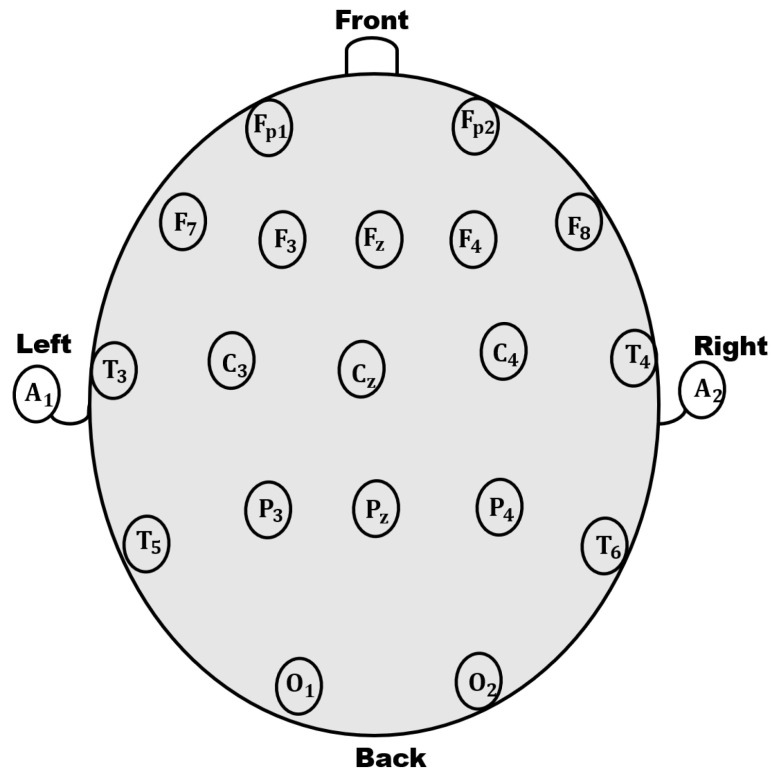
The EEG system with 10–20 international electrode placement.

**Figure 6 sensors-22-03036-f006:**
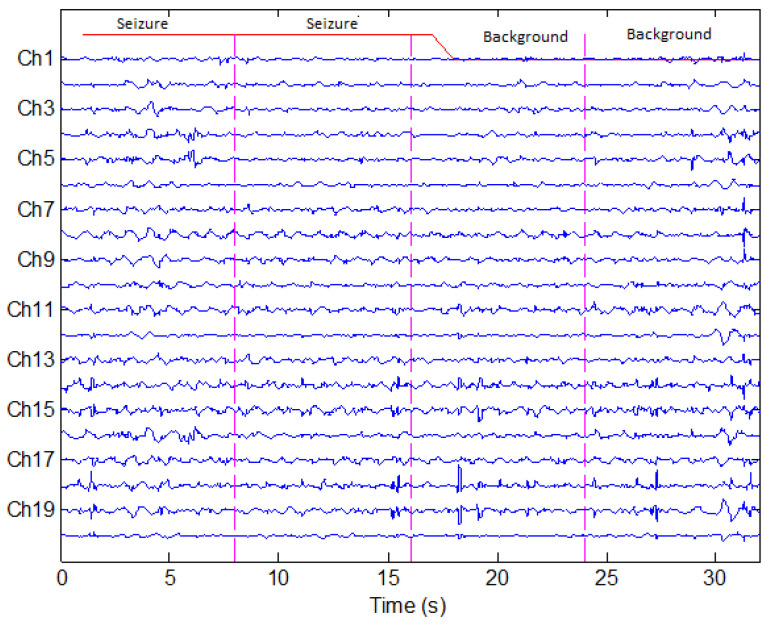
An example of the database showing seizure vs. background signals.

**Figure 7 sensors-22-03036-f007:**
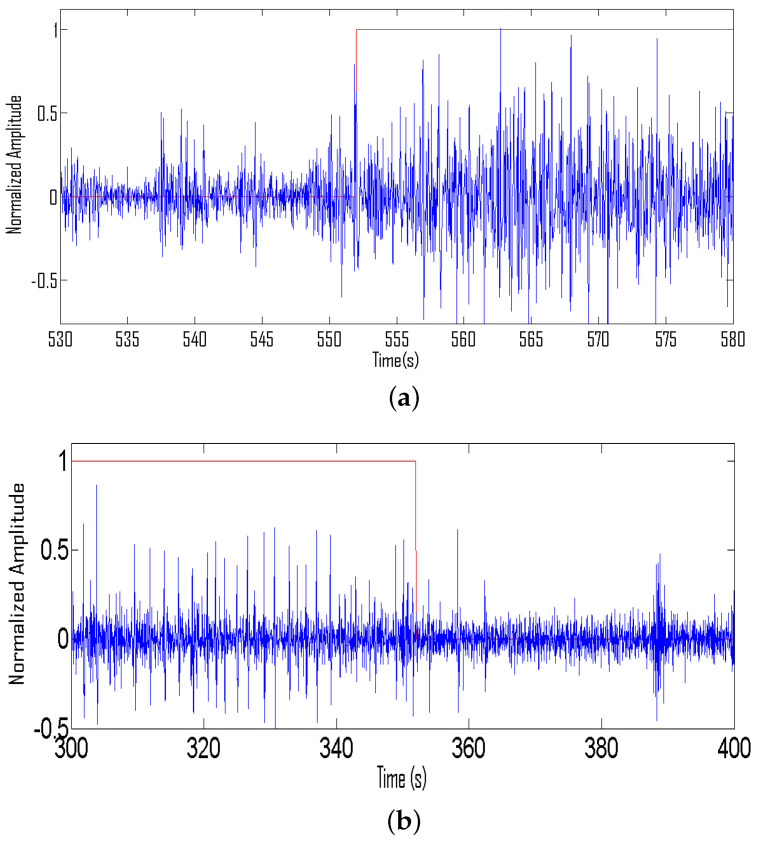
(**a**) Averaged and downsampled EEG signal just before and after the start of seizure activity, (**b**) Averaged and downsampled EEG signal just before and after the start of seizure activity.

**Figure 8 sensors-22-03036-f008:**
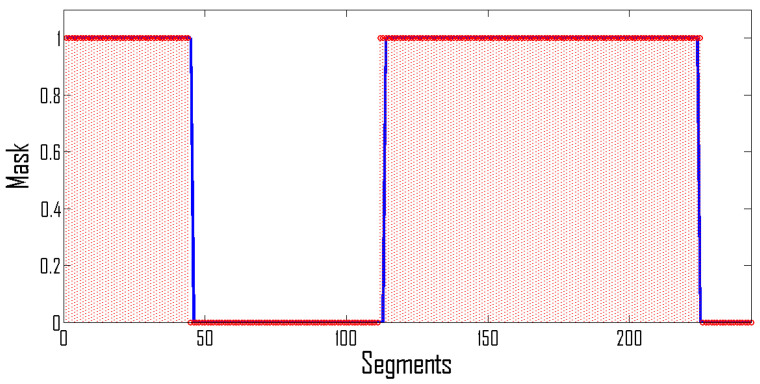
Original seizure mask (blue) vs. detected seizure (red).

**Figure 9 sensors-22-03036-f009:**
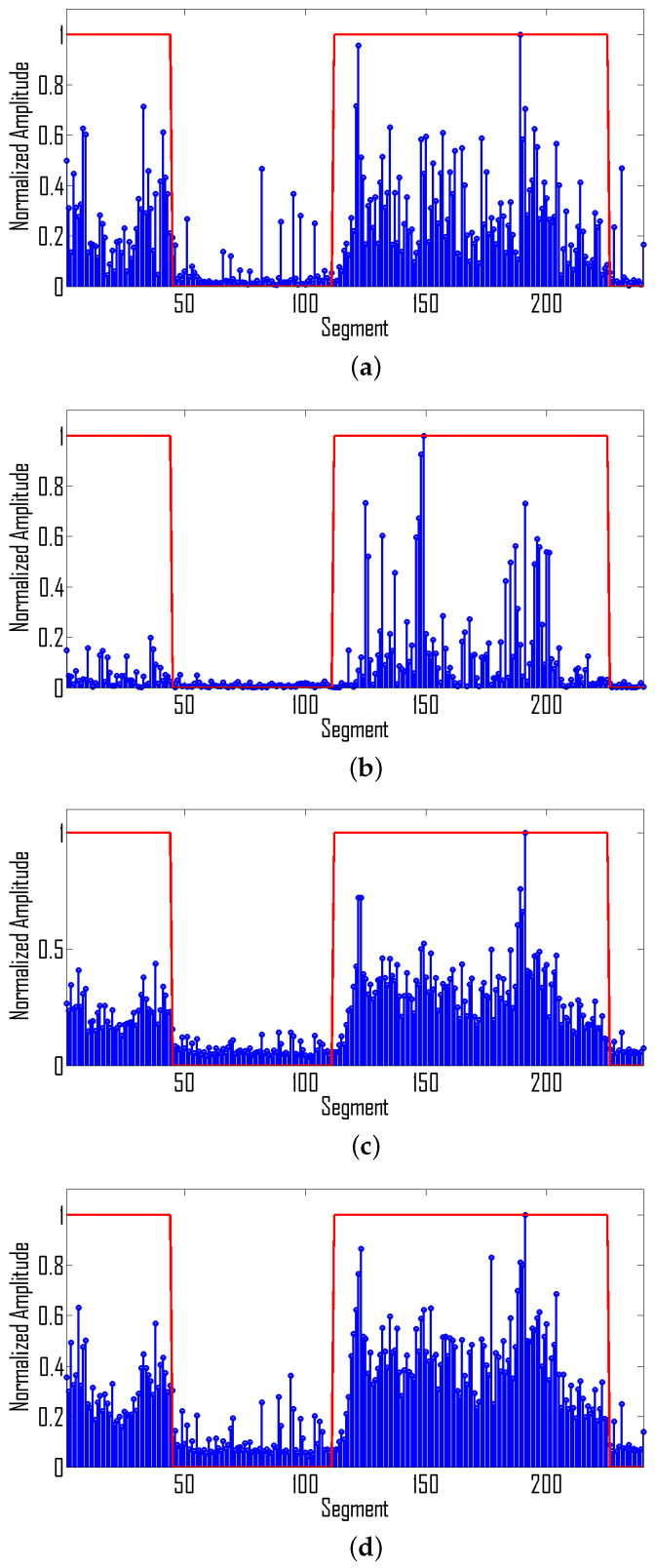
Illustration of feature variation during seizure activity: (**a**) Frequency marginal related feature, i.e., $F_{1}$, (**b**) Time-marginal related feature, i.e., $T_{1}$, (**c**) Mean of $\rho_{\mathrm{FIL}}\left(t,f\right)$ ($TF_{1}$), (**d**) standard deviation of $\rho_{\mathrm{FIL}}\left(t,f\right)$ ($TF_{2})$.

**Figure 10 sensors-22-03036-f010:**
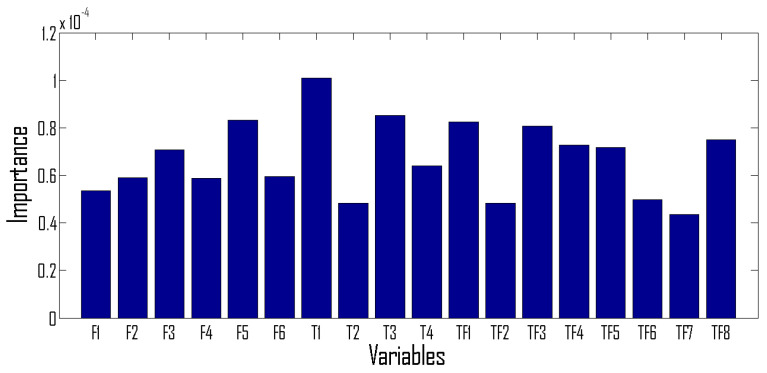
Importance of each feature. Note that $F_{1}$, $F_{2}$, $F_{3}$, $F_{4}$, and $F_{5}$ are modified frequency marginal features described in Table 1. $T_{1}$, $T_{2}$, $T_{3}$, and $T_{4}$ are modified time-marginal features described in Equation (2). $TF_{1}$, $TF_{2}$, …$TF_{8}$ are TF features defined in Table 2.

**Table 1 sensors-22-03036-t001:** Modified frequency marginal features.

S. No	Formula
1	$F_{1}=\int_{0}^{1}s\left(f\right)\mathrm{df}$
2	$F_{2}=\int_{1}^{2}s\left(f\right)\mathrm{df}$
3	$F_{3}=\int_{2}^{3}s\left(f\right)\mathrm{df}$
4	$F_{4}=\int_{3}^{4}s\left(f\right)\mathrm{df}$
5	$F_{5}=\int_{4}^{8}s\left(f\right)\mathrm{df}$
6	$F_{6}=\int_{8}^{16}s\left(f\right)\mathrm{df}$

**Table 2 sensors-22-03036-t002:** List of features along with their respective formula.

S. No	Name of Feature	Formula
1	Mean [9]	$TF_{1}=u=\int \int \rho_{\mathrm{FIL}}\left(t,f\right)dtdf$
2	Standard deviation [9]	$TF_{2}=\sigma^{2}=\int \int {\left(\rho_{\mathrm{FIL}}\left(t,f\right)-u\right)}^{2}\mathrm{dtdf}$
3	Co-efficient of variation [9]	$TF_{3}=\frac{\sigma }{u}$
4	Skewness [9]	$TF_{4}$=$\int \int {\left(\rho_{\mathrm{FIL}}\left(t,f\right)-u\right)}^{3}dtdf$
5	Kurtosis [9]	$TF_{5}$=$\int \int {\left(\rho_{\mathrm{FIL}}\left(t,f\right)-u\right)}^{4}dtdf$
6	Flatness [9]	$TF_{6}=\left(\frac{1}{u}\right)\prod_{t}\prod_{f}\rho_{\mathrm{FIL}}\left(t,f\right)$
7	Energy Concentration [9]	$TF_{7}=\int \int {\left|\rho_{\mathrm{FIL}}\left(t,f\right)\right|}^{1/2}\mathrm{dtdf}$
8	Flux [9]	$TF_{8}=\int \int \left|\frac{d}{\mathrm{dt}}\frac{d}{\mathrm{df}}\left(\rho_{\mathrm{FIL}}\left(t,f\right)\right)\right|\mathrm{dtdf}$

**Table 3 sensors-22-03036-t003:** Performance comparison using balanced dataset (i.e., equal examples of seizures and non-seizures).

Method	Sensitivity	Specificity	Accuracy	Precision	F1-Score
ADTFD	78.98%	84.07%	82.71%	64.36%	70.93%
CKD	76.01%	78.02%	77.48%	55.75%	64.32%
EMBD	75.24%	78.49%	77.62%	56.03%	64.23%
SPEC	76.01%	78.84%	78.08%	56.68%	64.94%

## Data Availability

We have used a multi-channel newborn EEG database that is described in [4,29,30] and was made publicly available along with relevant codes at https://github.com/ElsevierSoftwareX/SOFTX-D-17-00059 (accessed on 1 November 2019).

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
