# Peer review of "Modified Time-Frequency Marginal Features for Detection of Seizures in Newborns"

_sensors, 2022, doi:10.3390/s22083036_

Round 1
Reviewer 1 Report
Modified Time-frequency Marginal features for Detection of Seizures in Newborns
The authors have extracted a set of features from time and frequency domain of EEG signal to detect seizures in newborn.
Comments on abstract:
- It is highly recommended to provide some information on the patient population.
- The authors claim that their proposed analysis does a better job compared to existing algorithms. Putting a definite number helps readers to anticipate the results and remain interested.
- What kind of signal was used to do the seizure analysis.
- What is the utility of this method in clinical cases?
Comments:
- “Seizure detection algorithms usually involve the extraction of features from signal representation followed by training of a classifier signal”. This is a grossly misrepresented statement biased towards work only focused on machine learning based articles. [line 15]
- Methodology: Multichannel EEG signals are acquired and signals from all channels are averaged… The authors have not provided any details of the EEG signal, number of channels, sampling rate, montage configuration etc.
- Line 63: Provide brief details about the filter used.
- Figure 2 is not clear.
- What are the units on the Time signal for each a,b, and c.
- It appears the background is 100 times stronger than the seizure signals. Not sure about the representation of this signal. Is it accurate?
- The plots are 3D, it would be better to provide the unit of the 3rd dimension and the range of the values plotted.
- How many hours of background EEG was analyzed? And how many seizures were analyzed.
- What was the characteristic of the seizure that were analyzed? Figures of actual recorded EEG before averaging would be very helpful.
- How does the signal and features look like immediately before the seizure starts and after the seizure ends?
- Line 106: if seizure activity is partially present, was it considered seizure or non-seizure. How long the seizure activity must be there in an 8 sec. segment to be considered a seizure segment.
- There is no mention of who collected the data and IRB obtained to conduct this study. Without this information, the might be unethical.
- The random forest provides information on the rank of each feature corresponding to their contribution to the classification. The authors are requested to provide that information in the article.
- The results are not very convincing considering the amount of processing that is being done on the signal.
Author Response
Note: Graphics are not possible to paste here. Please see the attached word files.
Authors’ Response to Reviewers’ Comments
Manuscript ID: sensors-1643608
Type of manuscript: Article
Title: Modified Time-frequency Marginal features for Detection of Seizures in Newborns
Authors: Nabeel Khan, Sadiq Ali, Kwonhue Choi *
NOTE: The reviewer issues have been answered in the same order as they were formulated. The number of pages, references, equations and figures are always referred to the original draft unless otherwise specified.
We would like to thank the reviewer and editor for the effort and time devoted in the reviewing process. We sincerely appreciate the comments, the remarks on the original manuscript and the evaluation of the work. They have been very useful to further improve the clarity, completeness and argumentation of the paper. Below, we provide detailed answers to the required changes/modifications and explain how they have been taken into consideration in the revised manuscript.
Reviewer 1 Comments
Comments on abstract:
- It is highly recommended to provide some information on the patient population.
Ans: We fully understand this comment from the reviewer. The Data used in this work is already published. In this work, the aim was to test newly developed features for the detection of seizures in this publicly available data set. It is reported with the data that this data was acquired from 36 sick newborns admitted to the Royal Women’s Hospital, Brisbane, Australia [1][2].
[1] Sejdić, E., & Falk, T.H. (Eds.). (2018). Signal Processing and Machine Learning for Biomedical Big Data (1st ed.), Chapter 5. CRC Press. https://doi.org/10.1201/9781351061223
[2] Boualem Boashash, Samir Ouelha, “Designing high-resolution time–frequency and time–scale distributions for the analysis and classification of non-stationary signals: a tutorial review with a comparison of features performance”, Digital Signal Processing, Volume 77, 2018, Pages 120-152
- The authors claim that their proposed analysis does a better job compared to existing algorithms. Putting a definite number helps readers to anticipate the results and remain interested.
Ans: Thanks, we have now rephrased the sentence to bring more clarity. The proposed method achieves 5% better performance in terms of F1 score as compared to the existing method.
- What kind of signal was used to do the seizure analysis.
Ans: This study uses a large neonatal EEG database comprising multi-channel continuous EEG signals.
- What is the utility of this method in clinical cases?
Answer: We have now revised the abstract to highlight the utility of the proposed method in the clinical context.
Comments:
- “Seizure detection algorithms usually involve the extraction of features from signal representation followed by training of a classifier signal”. This is a grossly misrepresented statement biased towards work only focused on machine learning based articles. [line 15]
Ans: We fully understand the opinion of the reviewer. We have now rephrased the sentence as: “Machine learning-based seizure detection algorithms usually involve the extraction of features from signal representation followed by training of a classifier signal.”
- Methodology: Multichannel EEG signals are acquired and signals from all channels are averaged… The authors have not provided any details of the EEG signal, number of channels, sampling rate, montage configuration etc.
Ans: Thanks for bringing our attention to this important addition to the paper. We have provided such details in the result sections. We agree with the reviewer that more details are needed about the data. The Data is already published and publicly available. This neonatal EEG database comprises 20-channel continuous EEG signals. The data were recorded according to the 10–20 international electrode placement system using bipolar montage, using a Medelec Profile system (Medelec, Oxford Instruments, Old Woking, UK), and sampled at f s = 256Hz.
In the revised manuscript more details have been provided to avoid any confusion for the readers.
- Line 63: Provide brief details about the filter used.
Ans: Done as suggested. The differentiator filter is given as: . In the case of discrete implementation, it can be implemented as: .
- Figure 2 is not clear.
- What are the units on the Time signal for each a,b, and c.
- It appears the background is 100 times stronger than the seizure signals. Not sure about the representation of this signal. Is it accurate?
- The plots are 3D, it would be better to provide the unit of the 3rd dimension and the range of the values plotted.
Ans: We have provided 3D plots in Figure 2 for clarity as suggested.
- How many hours of background EEG was analyzed? And how many seizures were analyzed?
Ans: The data was acquired from 36 sick newborns. The database was annotated for seizure, artifacts, and normal background patterns. The durations of background, seizures, and artifacts are 6 hours, 5.16 hours, and 5.04 hours respectively. EEG signals are segmented into non-overlapping segments of 8-s. This database contains a total of 7290 segments including 2332 segments of seizures. This detail is now added to the manuscript in section 3.1.
- What was the characteristic of the seizure that were analyzed? Figures of actual recorded EEG before averaging would be very helpful.
Ans: In the revised manuscript we have provided a Snapshot of a newborn EEG showing seizure activity as shown below. Please see Figure 6 in the revised manuscript.
- How does the signal and features look like immediately before the seizure starts and after the seizure ends?
Ans: In the revised manuscript, Figure 7 illustrates EEG recording just before and just after seizure activity. Figure 9 illustrates features before the start of seizure activity and after the end of seizure activity is added. Whereas, an illustration of detected vs. original seizure using the balanced dataset is given in Figure 8.
- Line 106: if seizure activity is partially present, was it considered seizure or non-seizure. How long the seizure activity must be there in an 8 sec. segment to be considered a seizure segment.
Ans. We have used a publicly available dataset that is published in multiple publications i.e., [1][2]. The recorded dataset was divided into non-overlapping segments of duration 8-s and each segment was labeled as 1 if seizure activity was observed and 0 if no seizure activity was observed.
[1] Sejdić, E., & Falk, T.H. (Eds.). (2018). Signal Processing and Machine Learning for Biomedical Big Data (1st ed.), Chapter 5. CRC Press. https://doi.org/10.1201/9781351061223
[2] Boualem Boashash, Samir Ouelha, “Designing high-resolution time–frequency and time–scale distributions for the analysis and classification of non-stationary signals: a tutorial review with a comparison of features performance”, Digital Signal Processing, Volume 77, 2018, Pages 120-152
- There is no mention of who collected the data and IRB obtained to conduct this study. Without this information, the might be unethical.
Ans. The data used to assess our proposed signal processing mechanism was acquired from 36 sick newborns admitted to the Royal Women’s Hospital, Brisbane, Australia. The database was annotated forseizure, artifacts, and normal background patterns by a pediatricneurologist from the Royal Children’s Hospital, Brisbane, Australia. Ethical approval was given by Royal Brisbane and Women’s Hospital (RBWH), MedicalResearch Ethics Committee (MREC), Brisbane, Australia (ref:2005000095, dated 02/06/2005) [1].
[1] Khlif, M. S., P. B. Colditz, and B. Boashash. "Effective implementation of time–frequency matched filter with adapted pre and postprocessing for data-dependent detection of newborn seizures." Medical Engineering & Physics 35.12 (2013): 1762-1769.
- The random forest provides information on the rank of each feature corresponding to their contribution to the classification. The authors are requested to provide that information in the article.
Ans: We have provided such information. We are thankful to the reviewer for suggesting such valuable addition. Please see Figure 10.
- The results are not very convincing considering the amount of processing that is being done on the signal.
Ans: We believe that results should be compared with recent relevant research work. We have shown improvement in comparison to TF feature extraction-based approach developed in a recent study.
[1] Boualem Boashash, Samir Ouelha, “Designing high-resolution time-frequency and time–scale distributions for the analysis and classification of non-stationary signals: a tutorial review with a comparison of features performance”, Digital Signal Processing, Volume 77, 2018, Pages 120-152

Reviewer 2 Report
The authors reported that the time frequency signature of seizures spike and frequency-modulated chirps were able to find ictal discharges on scalp EEG more accurately than previous reported methods in neonates with epilepsy. This method seems to be used the reasonable features for detecting ictal discharges. However, some questions need to be resolved.
- The purpose of this study was to detect ictal discharges in patients with epilepsy, but the rationale for using neonatal EEG was unclear. The EEG of neonates shows generally slow-wave dominant, and the basic rhythms are not stable. In addition, seizures that begin in the neonatal period have various seizure semiology such as epileptic spasms and myoclonies. Many cases are severe epileptic encephalopathies with a variety of interictal and ictal findings on EEG. The reviewer think that neonatal EEG was not suitable for seizure detection. Please explain the reasons for using neonatal EEG as the subject of this study. What kind of neonatal seizure semiology and type of ictal discharges did the authors select for this study?
- The use of above EEG features for ictal discharges detection seems to be one of the rational choices. Please show us how many epochs did the authors used from scalp EEG of 36 cases, averaged 27 minutes, how many epochs were annotated as containing ictal or interictal discharges, respectively. Also, tell us what the authors treated the epochs containing artifacts. A reviewer was afraid of contaminating muscle artifacts during middle and later phase of seizure. If the authors used full length of ictal discharges for their detection, the results could be worsened. Please describe what considerations the authors have made.
- What was the reason for setting the segment as 8 seconds? If their purpose of this study was to find EEG changes in the seizure-onset, a shorter segment would be appropriate, whereas a longer segment would be more advantageous if the purpose was to detect overall changes in seizures.
- Many previous reports have presented that elevating power of the gamma and high gamma frequency bands were shown during ictal phase using scalp EEG. Why did the authors use down sampling to 32Hz? Even if this study could not show useful results for seizure detection in the gamma and high gamma bands, it was important to present them to compare with the previous papers.
- Emami et al. detected seizure on EEG using snap-shot EEG imaging and showed higher AUC through the CNN. (Neuroimage Clin. 2019;22:101684. doi: 10.1016/j.nicl.2019.101684.) The commercial based Persist softwareⓇ also can detect seizure using EEG signal with higher accuracy. What are the advantages using the author’s set of features? The clinical and/or academic physiological contribution of their results in this study should be considered.
- Why the authors extended the seizure boundary by 8-seconds at both ends once the seizure segment was detected? In order to use this method, end to end, it seems necessary to assume that edge control is not performed.
- The authors are necessary to show the EEG waveforms from random forest using their feature extraction.
- When the authors used 10-20 international EEG system, usually 16 electrodes for EEG analysis (if they included Fz, Cz, Pz for their analysis, it would be 19 electrodes). Please explain the detailed electrodes information the authors used.
- Was the EEG annotation performed by epileptologists or clinical EEG specialists?
Author Response
Noted: Graphics are not possible to paste here so please check the attached word file for this.
Authors’ Response to Reviewers’ Comments
Manuscript ID: sensors-1643608
Type of manuscript: Article
Title: Modified Time-frequency Marginal features for Detection of Seizures in Newborns
Authors: Nabeel Khan, Sadiq Ali, Kwonhue Choi *
NOTE: The reviewer issues have been answered in the same order as they were formulated. The number of pages, references, equations and figures are always referred to the original draft unless otherwise specified.
We would like to thank the reviewer and editor for the effort and time devoted to the reviewing process. We sincerely appreciate the comments, the remarks on the original manuscript, and the evaluation of the work. They have been very useful to further improve the clarity, completeness and argumentation of the paper. Below, we provide detailed answers to the required changes/modifications and explain how they have been taken into consideration in the revised manuscript.
Reviewer 2 Comments
The authors reported that the time frequency signature of seizures spike and frequency-modulated chirps were able to find ictal discharges on scalp EEG more accurately than previous reported methods in neonates with epilepsy. This method seems to be used the reasonable features for detecting ictal discharges. However, some questions need to be resolved.
Ans: Thanks to the reviewer for giving your valuable time to review our paper. We hope that response to reviewer comments will provide more value to the work reported in this paper.
- The purpose of this study was to detect ictal discharges in patients with epilepsy, but the rationale for using neonatal EEG was unclear. The EEG of neonates shows generally slow-wave dominant, and the basic rhythms are not stable. In addition, seizures that begin in the neonatal period have various seizure semiology such as epileptic spasms and myoclonies. Many cases are severe epileptic encephalopathies with a variety of interictal and ictal findings on EEG. The reviewer think that neonatal EEG was not suitable for seizure detection. Please explain the reasons for using neonatal EEG as the subject of this study. What kind of neonatal seizure semiology and type of ictal discharges did the authors select for this study?
Ans:
- Detection of seizures in newborns using EEG is an important problem as clinical signs are difficult to assess. So, timely detection of seizures can help save lives. This is highlighted in the abstract and introduction.
- The focus of this study has been to identify features in the joint TF domain for the accurate detection of seizures. In this regard, we have mainly developed an algorithm that considers seizures that appear as a train of spikes or seizures that are represented as frequency-modulated chirps. These two types of seizures are focused in this study are discussed in the introduction [1].
[1] “Tapani, Karoliina T., Sampsa Vanhatalo, and Nathan J. Stevenson. "Time-varying EEG correlations improve automated neonatal seizure detection." International journal of neural systems 29.04 (2019): 1850030.”
Comment#2: The use of above EEG features for ictal discharges detection seems to be one of the rational choices. Please show us how many epochs did the authors used from scalp EEG of 36 cases, averaged 27 minutes, how many epochs were annotated as containing ictal or interictal discharges, respectively. Also, tell us what the authors treated the epochs containing artifacts. A reviewer was afraid of contaminating muscle artifacts during middle and later phase of seizure. If the authors used full length of ictal discharges for their detection, the results could be worsened. Please describe what considerations the authors have made.
Ans. The database was acquired from 36 newborns using a 10--20 international electrode placement system. The signals were acquired at Royal Women's Hospital Brisbane. The database was annotated for seizure, artifacts, and normal background patterns. The durations of background, seizures, and artifacts are 6 hours, 5.16 hours, and 5.04 hours respectively. EEG signals are segmented into non-overlapping segments of 8-s. The recorded dataset was divided into non-overlapping segments of duration 8-s. This database contains a total of 7290 segments including 2332 segments of seizures. In this study, we have used all the segments of seizures and randomly selected segments of background and artifacts for the training of a classifier. In the future study, we aim to improve the performance of seizure detection by developing and integrating artifact detection and removal system.
The above information is added in the Experimental results and Conclusion sections.
- What was the reason for setting the segment as 8 seconds? If their purpose of this study was to find EEG changes in the seizure-onset, a shorter segment would be appropriate, whereas a longer segment would be more advantageous if the purpose was to detect overall changes in seizures.
Ans. We have used a publicly available dataset that was annotated with an 8-second duration. The focus of this study is seizure detection. So, we agree that larger segments can achieve better performance. This is now indicated as part of future work.
- Many previous reports have presented that elevating power of the gamma and high gamma frequency bands were shown during ictal phase using scalp EEG. Why did the authors use down sampling to 32Hz? Even if this study could not show useful results for seizure detection in the gamma and high gamma bands, it was important to present them to compare with the previous papers.
Ans. The previous study on the same dataset has shown that most of the signal energy of seizures is below 13 Hz. It was reported in the following study that approximately 78% of seizure activity takes place below 4 Hz, 15 % in between 4 Hz to 8 Hz, and 5.5 % from 8Hz to 12 Hz. Since the bulk of seizure activity is in the lower frequency bands, so we believe that it is appropriate to downsample the EEG data to 32 Hz.
This detail is now added to the paper in Section 3.1.
- Emami et al. detected seizure on EEG using snap-shot EEG imaging and showed higher AUC through the CNN. (Neuroimage Clin. 2019;22:101684. doi: 10.1016/j.nicl.2019.101684.) The commercial based Persist softwareⓇ also can detect seizure using EEG signal with higher accuracy. What are the advantages using the author’s set of features? The clinical and/or academic physiological contribution of their results in this study should be considered.
Ans. We believe that the paper mentioned by the reviewer is considering a different approach to detect seizures as the authors have considered images to extract features by CNN. Furthermore, CNN needs large datasets, as well as the mentioned work, which involves analysis of adult EEG whereas in our work EEG signals of newborns are considered. The focus of our proposed work is to develop new robust features that are effective in newborn EEG signals for machine learning algorithms. No doubt that neural network approaches are effective in large data sets but with limited data sets, robust feature-based machine learning techniques are still under research. Because the deep learning scheme assumes the availability of large EEG data sets. Large-scale data collection is still expensive, time-consuming, and restricted to a small number of teams working mainly in research laboratories [1].
As different sizes and the nature of databases are considered, the comparison is not straightforward. We believe that the comparison is done with recent relevant state of art and the proposed scheme outperforms.
[1] https://www.bitbrain.com/blog/ai-eeg-data-processing
- Why the authors extended the seizure boundary by 8-seconds at both ends once the seizure segment was detected? In order to use this method, end to end, it seems necessary to assume that edge control is not performed.
Ans. In order to reduce false positives, we first apply a 5-tap median filter, i.e., a given segment is classified as a seizure segment if at least a seizure appears in 3 out of 5 segments. Application of this median filter removes the seizure appearing at the boundaries thus reducing the overall sensitivity of the system. To improve the sensitivity, we extend the boundary by 8-s at each end. This detail is now added in the paper in Section 3.2.
- The authors are necessary to show the EEG waveforms from random forest using their feature extraction.
Ans: A figure has been added in the result section that shows the results of the proposed detector after the application of the random forest classifier vs the original seizure mask. Please see Figure 8.
- When the authors used 10-20 international EEG system, usually 16 electrodes for EEG analysis (if they included Fz, Cz, Pz for their analysis, it would be 19 electrodes). Please explain the detailed electrodes information the authors used.
Ans: We have used Fz, Cz, Pz for analysis. The configuration is illustrated in Figure 1. Each point on this figure to the left indicates possible positions of the electrode. For each location first letter identifies the lobe whereas the second number or letter identifies the hemisphere location. The letters F, T, C, P, and O stand for Frontal, Temporal, Central, Parietal and Occipital. Right hemisphere is represented by even numbers whereas odd numbers represent the left hemisphere. The z refers to an electrode placed on the midline [1][2]. This Figure along with the relevant detail is added in the paper in Section 3.1.
Figure: 1
1] Sejdić, E., & Falk, T.H. (Eds.). (2018). Signal Processing and Machine Learning for Biomedical Big Data (1st ed.), Chapter 5. CRC Press. https://doi.org/10.1201/9781351061223
[2] Boualem Boashash, Samir Ouelha, “Designing high-resolution time-frequency and time–scale distributions for the analysis and classification of non-stationary signals: a tutorial review with a comparison of features performance”, Digital Signal Processing, Volume 77, 2018, Pages 120-152
- Was the EEG annotation performed by epileptologists or clinical EEG specialists?
Ans: The database was annotated for seizure, artifacts, and normal background patterns by an expert pediatric neurologist from the Royal Children’s Hospital, Brisbane, Australia [1].
[1] Sejdić, E., & Falk, T.H. (Eds.). (2018). Signal Processing and Machine Learning for Biomedical Big Data (1st ed.), Chapter 5. CRC Press. https://doi.org/10.1201/9781351061223

Round 2
Reviewer 1 Report
The authors have addressed most of the review comments.
There are a few minor concerns:
- Figure 2:please ensure that the units are consistent across all plots for a fair comparison.
- Figure6: when showing an EEG segment, point the readers to which segments are seizure vs which segments are background.
- Figure 10: It is not clear what the F numbers mean. The authors had mentioned a list of features in the methods and the expectation is to rank them as per their relevance in predicting the seizures as per the training algorithm.
Author Response
Authors’ Response to Reviewers’ Comments
Manuscript ID: sensors-1643608
Type of manuscript: Article
Title: Modified Time-frequency Marginal features for Detection of Seizures in Newborns
Authors: Nabeel Khan, Sadiq Ali, Kwonhue Choi *
NOTE: The reviewer issues have been answered in the same order as they were formulated. The number of pages, references, equations and figures are always referred to the original draft unless otherwise specified.
We would like to thank the reviewer and editor for the effort and time devoted in the reviewing process. We sincerely appreciate the comments, the remarks on the original manuscript and the evaluation of the work. They have been very useful to further improve the clarity, completeness and argumentation of the paper. Below, we provide detailed answers to the required changes/modifications and explain how they have been taken into consideration in the revised manuscript.
Reviewer 1 Comments
Comments on abstract:
- Figure 2:please ensure that the units are consistent across all plots for a fair comparison.
Ans: Done as suggested.
- Figure6: when showing an EEG segment, point the readers to which segments are seizures vs which segments are background.
Ans. Done as suggested. The seizure and non-seizure part has been mentioned in the figure.
- Figure 10: It is not clear what the F numbers mean. The authors had mentioned a list of features in the methods and the expectation is to rank them as per their relevance in predicting the seizures as per the training algorithm.
Ans: Actually F1, F2, F3, F4, and F5 are the modified frequency marginal features described in Table 1. T1, T2, T3, and T4 are time-marginal features described in Equation 2. TF1, TF2, …TF8 are time-frequency features defined in Table 2. For avoiding any confusion for the readers, in the revised manuscript details are added in the caption Figure 10.

Reviewer 2 Report
The authors revised appropriately.
I think detection of seizure discharges are still challenging matter for the EMU, and detecting neonatal seizure is more difficult than adult seizure. Fundamental approach is to establish the machine learning based seizure detection algorism using common seizures occurred in adult patients with epilepsy. Subsequently, the researcher had better to apply the difficult problems, such as neonatal seizure.
However, I understand the purpose of this study was to detect the seizures in neonate to save their life. These were ones of the results in this field of research. The explanations for study design were improved to understand.
I will accept this paper for publication.
Author Response
Authors’ Response to Reviewers’ Comments
Manuscript ID: sensors-1643608
Type of manuscript: Article
Title: Modified Time-frequency Marginal features for Detection of Seizures in Newborns
Authors: Nabeel Khan, Sadiq Ali, Kwonhue Choi *
NOTE: The reviewer issues have been answered in the same order as they were formulated. The number of pages, references, equations and figures are always referred to the original draft unless otherwise specified.
We would like to thank the reviewer and editor for the effort and time devoted in the reviewing process. We sincerely appreciate the comments, the remarks on the original manuscript and the evaluation of the work. They have been very useful to further improve the clarity, completeness and argumentation of the paper. Below, we provide detailed answers to the required changes/modifications and explain how they have been taken into consideration in the revised manuscript.
Reviewer 2 Comments
I understand the purpose of this study was to detect the seizures in neonate to save their life. These were ones of the results in this field of research. The explanations for study design were improved to understand.
I will accept this paper for publication.
Ans: We are thankful to the reviewer for comprehensive comments in the previous iteration. They were very helpful in improving the paper.